# Genistein Modified with 8-Prenyl Group Suppresses Osteoclast Activity Directly via Its Prototype but Not Metabolite by Gut Microbiota

**DOI:** 10.3390/molecules27227811

**Published:** 2022-11-13

**Authors:** Zuo-Cheng Qiu, Feng-Xiang Zhang, Xue-Ling Hu, Yang-Yang Zhang, Zi-Ling Tang, Jie Zhang, Li Yang, Man-Sau Wong, Jia-Xu Chen, Hui-Hui Xiao

**Affiliations:** 1Guangzhou Key Laboratory of Formula-Pattern of Traditional Chinese Medicine, School of Traditional Chinese Medicine, Jinan University, Guangzhou 510632, China; 2Guangdong Provincial Key Laboratory of Traditional Chinese Medicine Informatization, Jinan University, Guangzhou 510632, China; 3State Key Laboratory for Chemistry and Molecular Engineering of Medicinal Resources, School of Chemistry and Pharmacy, Guangxi Normal University, Guilin 541004, China; 4College of Pharmacy, Jinan University, Guangzhou 510632, China; 5State Key Laboratory of Chinese Medicine and Molecular Pharmacology (Incubation), Shenzhen Research Institute of the Hong Kong Polytechnic University, Shenzhen 518057, China; 6Research Center for Chinese Medicine Innovation, The Hong Kong Polytechnic University, Hong Kong 999077, China

**Keywords:** 8-prenylgenistein, genistein, osteoclasts, gut microbiota, biotransformation

## Abstract

Postmenopausal osteoporosis is a significant threat to human health globally. Genistein, a soy-derived isoflavone, is regarded as a promising anti-osteoporosis drug with the effects of promoting osteoblastogenesis and suppressing osteoclastogenesis. However, its oral bioavailability (6.8%) is limited by water solubility, intestinal permeability, and biotransformation. Fortunately, 8-prenelylated genistein (8PG), a derivative of genistein found in Erythrina Variegate, presented excellent predicted oral bioavailability (51.64%) with an improved osteoblastogenesis effect, although its effects on osteoclastogenesis and intestinal biotransformation were still unclear. In this study, an in vitro microbial transformation platform and UPLC-QTOF/MS analysis method were developed to explore the functional metabolites of 8PG. RANKL-induced RAW264.7 cells were utilized to evaluate the effects of 8PG on osteoclastogenesis. Our results showed that genistein was transformed into dihydrogenistein and 5-hydroxy equol, while 8PG metabolites were undetectable under the same conditions. The 8PG (10^−6^ M) was more potent in inhibiting osteoclastogenesis than genistein (10^−5^ M) and it down-regulated NFATC1, cSRC, MMP-9 and Cathepsin K. It was concluded that 8-prenyl plays an important role in influencing the osteoclast activity and intestinal biotransformation of 8PG, which provides evidence supporting the further development of 8PG as a good anti-osteoporosis agent.

## 1. Introduction

Osteoporosis, especially postmenopausal osteoporosis, receives worldwide attention since it is a significant threat to human health. It is reported that approximately 10% of the world’s population and more than 30% of postmenopausal women aged over 50 years suffer from osteoporosis [1,2,3]. Osteoporosis is caused by a dramatic imbalance in bone remodeling; in other words, osteoclast-involved bone resorption outweighs osteoblast-involved bone formation [4]. Notably, osteoclast differentiation, a part of osteoclast-involved bone resorption, is regulated by a variety of factors, such as receptor activator of nuclear factor (NF)-κB ligand (RANKL)—the major osteoclastogenic molecule [5]. The binding of RANKL to the receptor activator of nuclear factor (NF)-κB (RANK) leads to the recruitment of tumor necrosis factor receptor-associated factor 6 (TRAF6), triggering the activation of transcription factors, such as the nuclear factor of activated T-cells (NFAT) [6]. Excessive osteoclast differentiation can result in osteoporosis, indicating that suppressing osteoclast activity is very important to treat osteoporosis.

Hormone-replacement therapy (HRT) was once regarded as the most effective therapy for postmenopausal osteoporosis [7]. However, hormone therapy for the prevention of osteoporosis has been questioned due to safety concerns [8,9], and it has been restricted since increased side effects caused by long-term use were found, such as breast cancer and uterine cancer [10]. Recently, phytoestrogens (especially in foods), with excellent anti-osteoporosis effects and lower side effects, were widely treated as a potential alternative therapy. Genistein, a soy-derived phytoestrogen with the ability to increase osteoblastogenesis and suppress osteoclastogenesis, has attracted worldwide attention, and its anti-osteoporosis effects have been widely evaluated using animal models [11] or clinical trials [12]. However, it still has some limitations given that more than 80% of genistein is converted to its glucuronides and sulfates [13], and the absolute bioavailability of genistein is 6.8% [14]. Thus, there is an urgent need to explore candidates with excellent anti-osteoporosis effects and improved oral bioavailability.

As reported in previous work, 8-prenelylated genistein (8PG), a derivative of genistein (G) in *Erythrina variegate* (EV) extracts, exerts protective effects against bone loss in ovariectomized mice [15]. In addition to its strong chemical similarity with genistein, 8PG exerted an improved anti-adipogenesis effect with a lower dosage and improved ADME properties (predicted oral bioavailability 51.64%) [4]. Meanwhile, the function of 8PG in osteogenesis was evaluated [4], but its effects and mechanism on osteoclast activity were not reported. Notably, oral administration is the primary form of drug access for patients, and gut microbiota dramatically influence drug absorption, especially for long-term-use drugs [16]. The metabolic evaluation of the drug demonstrated its great importance. Currently, the metabolic study of 8PG and its effects on osteoclast activity is still unclear. Meanwhile, whether 8PG has a similar anti-osteoporosis mechanism and functional formation to genistein or not also remains unelucidated. In this work, an in vitro microbiota model and RANKL-induced RAW264.7 cell model was developed to explore their functional format and the mechanism of 8PG and genistein on osteoclastogenesis, and their difference is discussed.

## 2. Results

### 2.1. Mass-Fragmentation Behaviors of 8-Prenelylated Genistein and Genistein

The mass fragmentations of 8-prenelylated genistein and genistein are listed in Figure 1A. The 8-prenelylated genistein (8PG) presented a parent deprotonated ion at *m*/*z* 337.1075 (M-H)^−^. In the MS/MS spectrometry, it produced three primary ions at *m*/*z* 282.0528 (M-H-C_4_H_7_)^−^, 263.0501 (M-H-C_4_H_7_-CO)^−^, and 225.0553 (M-H-C_4_H_7_-CO)^−^. The genistein showed a parent ion at *m*/*z* 269.0445 (M-H)^−^ and two primary ions at *m*/*z* 241.0486 (M-H-CO)^−^ and 213.0557 (M-H-2CO)^−^ (Figure 1B). It also produced a fragment ion at *m*/*z* 225.0545 (M-H-CO_2_)^−^. Interestingly, this was formed by losing the O of the C ring in the flavones, which was reported in a previous work [17]. The fragment ions at *m*/*z* 201.0543 (M-H-C_3_O_2_)^−^ and 159.0434 (M-H-C_3_O_2_-C_2_H_2_O)^−^ were produced by losing C_3_O_2_ and C_2_H_2_O. Furthermore, fragment ions, such as *m*/*z* 133.0279 (^0,3^B)^−^, can also be formed by Retro Diels-Alder reaction (RDA cleavage). 

### 2.2. Effect of Prenyl Group on the Metabolic Conversion of Isoflavonoids by Gut Bacteria

Prenyl-isoflavonoids are the major components in EV [18,19,20,21]. To determine whether the prenyl group would alter the metabolic conversion of isoflavonoid through the gut bacteria, the metabolism of the genistein and 8-prenylgenistein (a genistein derivative with the prenyl group at the 8-position naturally present in EV [19]) by the gut microbiota were studied using an in vitro incubation method. The representative ion chromatograms of blank incubated bacterial-culture media, blank incubated bacterial-culture media spiked with genistein, dihydrogenistein, 5-hydroxy equol and 8-prenylgenistein, and the incubated genistein and 8-prenyl genistein spiked with rutin (internal standard) are shown in Figure 2A–C. The incubation results showed that the genistein was first metabolized to dihydrogenistein at 8 h and further degraded to 5-hydroxy equol, which reached the highest concentration on the fifth day of incubation (Figure 3A,B). By contrast, no detectable metabolites of 8-prenylgenistein were identified in the incubated bacterial culture media throughout the incubation period (Figure 3A,B). The results suggested that the prenyl group might influence the metabolism of genistein derivatives.

### 2.3. PG Exhibited More Potent Activity Than Genistein in Suppressing RANKL-Induced Osteoclastogenesis, F-Actin Ring Formation, and Bone-Resorption Activity

The effects of the 8PG and genistein on the pre-osteoclast viability and the osteoclast differentiation and function in RAW 264.7 cells were determined by CCK-8 assay, tartrate resistance acid phosphatase (TRAP) staining on multinucleated cells (MNCs, TRAP-positive MNCs were counted as osteoclast cells), F-actin ring formation, and bone resorption assay, respectively. The 8PG and genistein (10^−6^ to 10^−5^ M) did not exert cytotoxic effects on the pre-osteoclast RAW 264.7 cells (Figure 4A). The formation of MNCs in the Ctrl (+) group was significantly induced by RANKL (Figure 4B,C, *p* < 0.001). The E2 (10^−8^ M, 17-β estradiol, positive control), but not genistein, at the tested concentrations significantly (*p* < 0.05) reduced the number of TRAP-positive MNCs. The 8PG significantly suppressed the formation of TRAP-positive MNCs at 5 × 10^−6^ M (*p* < 0.01) to 10^−5^ M (*p* < 0.001). To determine the effect of the compounds on the osteoclast maturation, the F-actin ring formation was measured by phalloidin-Alexa Fluor 555 staining. As shown in Figure 5A, the F-actin ring shaped well in the RANKL-induced control group (Ctrl (+)), while the E2 and 8PG (10^−6^ to 10^−5^) potently disrupted the F-actin ring. By contrast, the genistein only slightly disrupted the F-actin ring structure at the high dosage (10^−5^ M). Furthermore, the effect of the compounds on the osteoclastic bone reportion was also assessed; the E2 and genistein attenuated the osteoclastic resorption activity at the tested concentrations (Figure 5B,C), and the 8PG almost completely abrogated the increase in pit area at 5 × 10^−6^ M (*p* < 0.001) and 10^−5^ M (*p* < 0.001). Additionally, the mRNA expression of the RANKL-induced osteoclast-associated genes, including c-SRC, NFATC1, Cathepsin K, and MMP-9 was significantly up-regulated in the Ctrl (+) group compared with the Ctrl (−). The up-regulation of these genes was attenuated more significantly by the 8PG than by the genistein at the same concentration (Figure 5D–G).

### 2.4. Predicted Anti-Mechanism of 8-Prenelylated Genistein and Genistein Based on Prototypes and Metabolites

The development of drug-originated metabolites was a major therapy achievement, and the prediction of the targets of metabolites is necessary to reveal their therapeutic mechanism [22,23]. Thus, the prototypes and metabolites were subjected to target prediction.

A total of 140 targets from 4 constituents (8PG, genistein, dihydrogenistein, and 5-hydroxy equol), with a probability of more than 0.1, were obtained using Swiss Target Prediction. Next, the compound-target network was constructed by using Cytoscape software. As shown in Figure 6A, the network consisted of 144 nodes and 216 interactions. Meanwhile, the osteoporosis-related targets, with ‘Score_gda’ > 0.1, were obtained in DisGeNet. The Venn diagram of genistein, 8PG, the metabolites of genistein, and osteoporosis are shown in Figure 6B. In total, 42 overlapping targets between 8PG and osteoporosis, genistein and osteoporosis, the metabolites of genistein, and osteoporosis were selected after the deletion of the duplicates. These 42 unique targets were regarded as the core targets for 8PG and genistein in the treatment of osteoporosis. The core targets, such as SRC, MMP9, MMP2 were related to the anti-osteoporosis effects of the genistein (Figure 6(C1)), while the AKT, SIRT1, RELA and so on were correlated with the effects of the 8PG on osteoporosis. The different core targets in response to the genistein and 8PG indicated that the actions of the two compounds on bone might take place different signaling pathways.

The KEGG-classification analyses of the target proteins and the pathways were performed using Metascape database. The results showed that the genistein was mainly involved in “response to hormones” (Figure 6(D1)), while the 8PG was involved in “serotoergic synapse” (Figure 6(D2)).

## 3. Discussion

Accumulating evidence suggests that the gut microbiota plays a significant role in the metabolism, bioavailability, and bioactivity of medicines, especially those that are orally administrated. The gut microbiota contains large quantities of various types of enzymes that transform medicines. Microbial enzymes catalyze various types of reactions, including oxidation, reduction, decarboxylation, demethylation, isomerization, and ring cleavage [24]. The conversion of isoflavonoids by intestinal bacteria has an impact on their biological effects; for example, the daidzein metabolite equol exhibits biological properties that exceed those of its precursor [25]. However, only 25–60% of adults excrete equol after soy consumption, which is believed to be dependent on the presence of equol-forming bacteria in the gut microflora. The intestinal microbiota responsible for equol production might differ across individuals. Bacteria such as *Adlercreutzia equolifaciens*, *Eggerthellia* spp. *Escherichia coli*, *Bacteroides ovatus*, *Ruminococcus productus*, *Streptococcus intermedius* and *Slackia isoflavoniconvertens* in humans, and *Enterohabdus*
*musicola* and *Asaccgaribater celatus* in mice [24,26], are reported to be involved in the transformation of daidzein and genistein to equol. Coldham et al. [27] compared the metabolism of genistein in human- and rat-gut microflora, demonstrating that the same metabolites of genistein were identified after incubation with rat caecal microflora and human faecal microflora, while the transformation times of these metabolites were different between the two species. The 8-prenylflavonoids are reported to have higher osteogenic activity than their parent flavonoids, but studies on the transformation of prenylflavonoids by gut microbiota are limited. Terao and Mukai proposed that prenylflavonoids are likely to be converted through similar metabolic pathways to flavonoids by microbiota [28]. In our present rat-fecal-microbiota transformation study, we found that the transformations of genistein and 8-prenylgenistein by gut microbiota were different (Figure 5): genistein, but not 8-prenylgenistein, was transformed to dihydrogenistein and then converted to 5-hydroxyl equol by the anaerobic incubation of the gut microbiota. The results suggested that the prenyl group might hinder the transformation of prenylflavonoids, which was in agreement with a previous study by others showing that the nonpolar methoxy groups on ring A or B of flavonoids could protect them from extensive bacterial degradation [29]. In addition, the prenylation of flavonoids lowered their plasma and lymph concentrations (as an index of intestinal absorption) but increased their accumulation by at least 10 times in the liver and kidneys when compared with the parent flavonoids [28,30], suggesting a beneficial role of the prenyl group in the tissue accumulation of prenylflavonoids. Thus, prenylflavonoids in EV extract are likely to be absorbed directly by intestinal cells without transformation by the gut microbiota and have high accumulation concentrations in tissues. The 8PG, but not the genistein, almost completely inhibited the osteoclastic differentiation and resorption at 5 × 10^−6^ M and 10^−5^ M.

A dramatic imbalance in osteoclast-involved bone resorption and osteoblast-involved bone formation can cause osteoporosis, including excessive osteoclast differentiation and reduced osteogenesis. This indicates that suppressing osteoclast activity is a core factor in bone health. Generally, NFATC1 was confirmed as a master regulator of the osteoclast transcriptome, promoting the expression of numerous genes needed for bone resorption [31]. As reported in previous work, the deletion of c-SRC in mice can alert the osteoclast with reduced bone resorption, leading to osteopetrosis [32]. Matrix metalloproteinase-9 (MMP-9) is specifically required for the invasion of osteoclasts and endothelial cells into the discontinuously mineralized hypertrophic cartilage that fills the core of the diaphysis. This indicates that MMP-9 deficiency exhibits a delay in osteoclast recruitment [33]. Cathepsin K (CTSK) is secreted by osteoclasts to degrade collagen and other matrix proteins during bone resorption. The global deletion of Ctsk in mice decreases bone resorption, leading to osteopetrosis, but also increases the bone-formation rate (BFR) [34]. This information suggests that NFATC1, cSRC, MMP-9 and Cathepsin K were the four core indicators of osteoclastogenesis involved in osteoporosis. The overexpression of NFATC1, cSRC, MMP-9, and Cathepsin K was found in RANKL-induced osteoclastogenesis, and 8PG significantly inhibited their expressions, indicating that the 8PG achieved anti-osteoclastogenesis by downregulating NFATC1, cSRC, MMP-9, and Cathepsin K. Despite having a similar chemical structure, the genistein achieved anti-osteoclastogenesis only through NFATC1 and cSRC. It had no significant influence on MMP-9 or Cathepsin K. These differences might have been caused by their functional format, since the genistein could be metabolized into its metabolies, while the 8PG was still presented as a prototype.

## 4. Materials and Methods

### 4.1. Materials

The 17β-Estradiol (purity ≥ 98%) and genistein (purity ≥ 98%) were purchased from Sigma-Aldrich (St. Louis, MO, USA). Minimum Essential Medium α (α-MEM), Dulbecco’s modified Eagle medium (DMEM), fetal bovine serum (FBS,), charcoal-stripped fetal bovine serum (sFBS), 0.5% Trypsin-EDTA and penicillin-streptomycin were the products of Gibco (Gaithersburg, MD, USA). Murine sRANKL and M-CSF were the products of Peprotech (Rocky Hill, NJ, USA). The 8-prenylgenistein was synthesized starting from genistein according to [35]; the detailed procedures are described in the Appendix A.

### 4.2. Animal Study

Sprague–Dawley specific-pathogen-free (SPF) female rats (*n* = 48) at 3 months old were obtained from Vital River Laboratory Animal Technology Company (Beijing, China). All rats were housed in the standard condition of 12 h light/12 h dark cycle, at a temperature of 20–22 °C and at 40~60% humidity. During the experiment, all rats were allowed free access to normal diet and distilled water. The animals’ welfare and experimental protocol were strictly in accordance with the procedures approved by the Animal Ethic Committee of the Hong Kong Polytechnic University (no. 190703).

### 4.3. Incubation of Anaerobic Microbiome with Compounds

Fecal samples were freshly collected from the rats after 7 days’ acclimatization. A ten-milliliter brain–heart infusion (BHI) was added into a fifteen-milliliter sterile tube and covered with a thin layer of paraffin oil to create an anaerobic condition. Fecal samples were mixed and inoculated into BHI broth and incubated at 37 °C overnight with mild shaking (50 rpm). Each 500-microliter aliquot of bacterial culture was mixed with 500 µL of BHI and 30% glycerol, covered with a thin layer of paraffin oil, and stored at −80 °C. The culture stock was thawed before incubation with compounds, in which a 200-microliter aliquot was inoculated into 10 mL of BHI broth, covered with paraffin oil, and incubated at 37 °C overnight with mild shaking (50 rpm). A total of 50 μL of overnight culture was inoculated into 5 mL BHI broth covered with a thin layer of paraffin oil and incubated at 37 °C for 3~4 h, after which a vehicle, 1 mg/mL of compound dissolved in 1% DMSO, was added into 2-milliliter aliquots of bacteria culture and incubated for 0 h, 8 h, 1 day, 2 days, 3 days, 4 days, 5 days, 6 days, and 7 days. The bacterial culture was centrifuged at 12,000 rpm for 2 min, and the supernatant was transferred into a clean tube for further liquid chromatography–mass spectrometry (LC–MS) analysis.

### 4.4. Sample Preparation and Ultra-Performance Liquid-Chromatography–Quadruple-Time-of-Flight–Mass Spectrometry (UPLC–QTOF–MS) Analysis

A total of 100 μL of thawed bacterial culture medium was mixed with 300 μL methanol (including 10 ppm rutin as internal standard), and the mixture was kept at 4 °C for 2 h for completion of deproteination. A total of 340 μL supernatant was collected after centrifugation at 18,700× *g* at 4 °C for 20 min and dried in nitrogen gas. The dried supernatant was reconstituted in 100 uL of methanol: water (50:50, *v*/*v*) prior to UPLC–QTOF–MS analysis.

The microbiome-metabolite profiling was performed with a Waters ACQUITY UPLC system coupled with Waters SYNAPT G2 Q-IM-TOF HDMS system (Waters, Milford, MA, USA). LC analysis was carried out with a Phenomenex Luna Omega Polar C18 column (50 × 2.1 mm, 1.6 μm). A 5-microliter aliquot was eluted by 0.1% formic acid in water (*v*/*v*, A) and 0.1% formic acid in acetonitrile (*v*/*v*, B) at a flow rate of 0.3 mL/min with elution gradient as follows: 0–2 min, 15% B; 7 min, 25% B; 12 min, 45% B; 14 min, 70% B; 16–18 min, 95% B. Column and sample chamber temperature were set at 40 °C and 6 °C, respectively. Mass spectrometry was performed in an electrospray ion source in negative-ionization mode. Nitrogen and argon were used as cone and collision gases. The MS parameters were set as follows: capillary voltage, 2.0 kV; sample-cone voltage, 30.0 V; extraction-cone voltage, 4.0 V; source temperature, 120 °C; desolvation temperature, 400 °C; gas flows of cone and desolvation, 50 and 800 L/h, respectively. The scan time was 0.5 s, with a 0.024 s interscan delay. A data-scan range from *m*/*z* 100 to 1500 was recorded. The MSE experiment was performed as follows: trap-collision energy of low energy function, 4.0 eV; ramp-trap collision energy of high energy function, 20–40 eV. In this section, argon was employed as collision gas. For accurate mass acquisition, a lock-mass of leucine encephalin was used along with 30 eV of trap collision energy, 2.5 kV of capillary energy, and 40 V of cone voltage; monitoring for negative-ion mode ((M-H)^−^: *m*/*z* 236.1035, 554.2615). All data were processed with Metabolynx XS software v 4.1 and Markerlynx application manager v 4.1 SCN 901 (Waters, Milford, MA, USA).

### 4.5. Cell Culture

Murine pre-osteoclastic RAW264.7 cells were purchased from ATCC (American Type Culture Collection, Manassas, VA, USA). The RAW264.7 cells were cultured in Dulbecco’s modified Eagle’s medium (DMEM) respectively, supplemented with 10% FBS and antibiotics (100 U/mL penicillin and 100 μg/mL streptomycin) in a humidified incubator with 95% air and 5% CO_2_ at 37 °C. Cells were digested with 0.5% Trypsin-EDTA and sub-cultured every two to three days.

### 4.6. Osteoclast Differentiation, F-Actin-Ring Formation and Resorption of RANKL-Induced RAW264.7 Cells

#### 4.6.1. Cell-Cytotoxicity Assay

RAW264.7 cells (ATCC, Manassas, VA, USA) were seeded at a cell density of 5000 cells/well in 96-well plate. The culture medium was phenol-red free alpha MEM (Invitrogen, Carlsbad CA, USA), supplemented with 10% sFBS, penicillin 100 U/mL and streptomycin 100 μg/mL. The cytotoxicities of cells upon treatment with Ctrl (1% ethanol), E2 (10^−8^ M), 8PG (10^−6^ to 10^−5^ M), and genistein (10^−6^ to 10^−5^ M) for 4 days were determined by using CCK-8 kit, according to the manufacturer’s instruction (Dojindo, Kumamoto, Japan).

#### 4.6.2. Osteoclast Differentiation

Cells were subsequently induced by differentiation medium (phenol-red free alpha MEM, 10% FBS, containing 30 ng/mL M-CSF, 100 ng/mL sRANKL) with Ctrl (1% ethanol), E2 (10^−8^ M), 8PG (10^−6^ to 10^−5^ M), and G (10^−6^ to 10^−5^ M) for 4 days. Cells were then fixed and stained for tartrate-resistant acid phosphatase (TRAP) using leukocyte-acid-phopshatase kit (sigma, USA). TRAP-positive cells showing more than three nuclei were counted as mature osteoclasts.

#### 4.6.3. Immunofluorescence Staining for F-Actin of Osteoclasts

The RAW264.7 cells were seeded at 3000 cells/well onto the Corning^®®^ Osteo Assay Surface 96-well multiple-well plate (Corning Incorporated Life Science, Corning, NY, USA). The RAW264.7 cells were treated with osteoclast-differentiation medium with the treatment of puerarin for 4 days. After the treatment, cells were rinsed with PBS 3 times and fixed in 4% paraformaldehyde for 30 min at 4 °C. Next, BMMs were permeabilized for 10 min in 0.1% X-100 in PBS. After a brief washing in PBS, cells were blocked with 5% bovine serum albumin for 1 h. Next, cells were incubated with Phalloidin-iFluor 555 (Abcam, Boston, MA, USA) for 2 h to label the F-actin ring. After treatment with actin-ring staining, cells were washed three times with PBS, followed by staining nuclei with DAPI (Beyotime Biotechnology, Shanghai, China) for 5 min; cell images were visualized and captured using fluorescence microscope (Zeiss vert.A1, Jena, Germany).

#### 4.6.4. Bone-Resorption Activity

The bone-resorption activity was also measured. Cells were seeded onto the Corning^®^ Osteo Assay Surface 96-well multiple-well plate (Corning Incorporated Life Science, Corning, NY, USA) at a density of 5 × 10^3^ cells/well. The cells were then attached to the bottom for 24 h before the medium was replaced with differentiation medium with different drug treatments. The cells were cultured on this plate for 6 days with medium changes every other day. Each experiment was performed on six separate wells. The cells were removed by treatment with 5% sodium hypochlorite for 5 min and the wells were washed with water and dried. The wells were imaged with microscopy (Olympus IX71, Hamamatsu-City, Japan to identify resorption pits. Areas of resorption pits were quantified using Image-Pro Plus v6.0 software (Media Cybernetics, Silver Spring, MD, USA).

### 4.7. Real-Time PCR

Total RNA was extracted using a RNeasy kit (Qiagen, Valencia, CA, USA) according to the manufacturer’s instructions. The reverse transcription was performed using PrimeScriptTM RT Master Mix (TaKaRa, Otsu, Shiga, Japan). The primer sequences used in the assay were listed in Appendix A. The 10 μL of the final reaction solution contained 1 μL of the diluted cDNA product, 5 μL of 2 × TB Green Premix Ex Taq II (TaKaRa, Otsu, Shiga, Japan), 0.5 μL each of forward and reverse primers, and 4 μL nuclease-free water. The amplification conditions and procedures were as follows: 50 °C for 2 min, 95 °C for 10 min, 40 cycles of 95 °C for 15 s, 60 °C for 1 min. The fluorescence signal was recorded by Roche Light Cycler 480 Detection System, after which the signal was converted into numerical values. Relative gene expression was determined by employing the Comparative CT method. The mRNA levels of all genes were normalized by β-actin as internal control. These analyses were performed in duplicate for each sample using cells from two different cultured wells, and each experiment was repeated three times.

### 4.8. Network Pharmacological Analysis

The targets (Homo species) of 8-prenelylated genistein and genistein and the metabolites of genistein were retrieved and collected from Swiss Target Prediction (http://www.swisstargetprediction.ch; accessed on 21 October 2022) [36] by importing the compounds with SMILES format. The osteoporosis-related targets were obtained from the DisGeNet database [37]. The protein–protein interactions (PPIs) were achieved by the STRING database (version 11.0, https://string-db.org/; accessed on 21 October 2022) [38], and protein interactions, with a confidence score > 0.7, were obtained after eliminating duplicates. The chemical-constituent-target networks and protein–protein interaction (PPI) networks were constructed and viewed by Cytoscape software (version 3.2.1) [39], and the network was analyzed by default setting with “degree” value. All proteins/genes were subjected to pathway-enrichment analysis (KEGG analysis) by using the Metascape (https://metascape.org/; accessed on 21 October 2022)—an online platform [40].

### 4.9. Statistical Analysis

Results are presented as mean ± SEM. Inter-group differences were determined by one-way analysis of variance (ANOVA), followed by post hoc Tukey’s test for multiple comparisons in PRISM version 5.0.1 (GraphPad, San Diego, CA, USA). Any *p*-value < 0.05 was considered as statistically significant.

## 5. Conclusions

Overall, this study demonstrated that 8-prenylgenistein was more potent than genistein in exerting bone-protecting effects, and the underlying mechanism might include both osteogenesis and osteoclasts. However, it should be noted that the present study only evaluated the in vitro anti-osteoclastogenic activities of 8PG, and an in vivo study is required to confirm the role of these signaling pathways in mediating the bone-protecting function of 8PG. In addition, the use of rat microbiota but not human microbiota for 8PG transformation is a limitation of the present study. A further metabolism study of 8PG by human microbiota will be needed. Our study systematically characterized the effects of 8-prenylgenistein on osteoclastogenesis and identified the potential biological targets involved in mediating the bone-protecting effects of 8PG. This study increased the understanding of the molecular actions of prenylated flavonoids in exerting bone-protecting effects and provided evidence to support their use in the management of bone health.

## Figures and Tables

**Figure 1 molecules-27-07811-f001:**
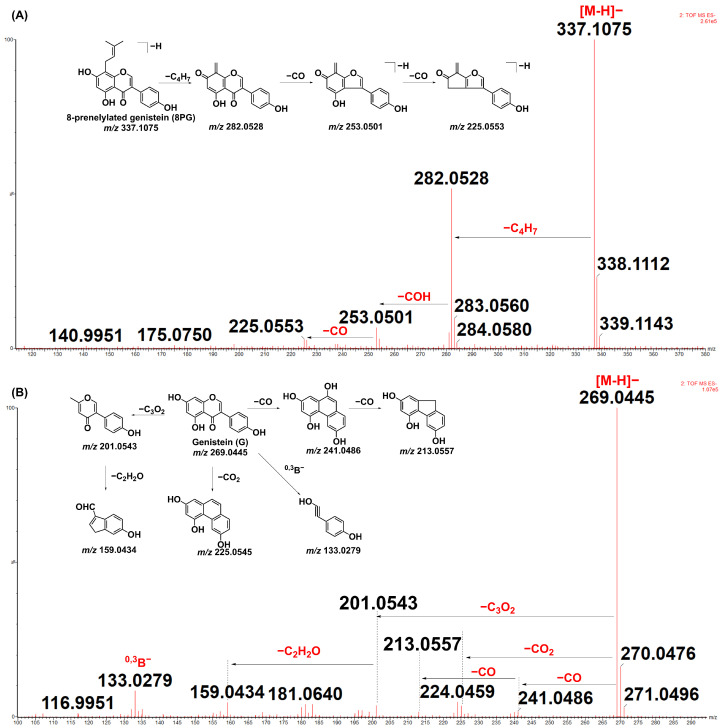
The mass-fragmentation behaviors of 8-prenelylated genistein (**A**) and genistein (**B**).

**Figure 2 molecules-27-07811-f002:**
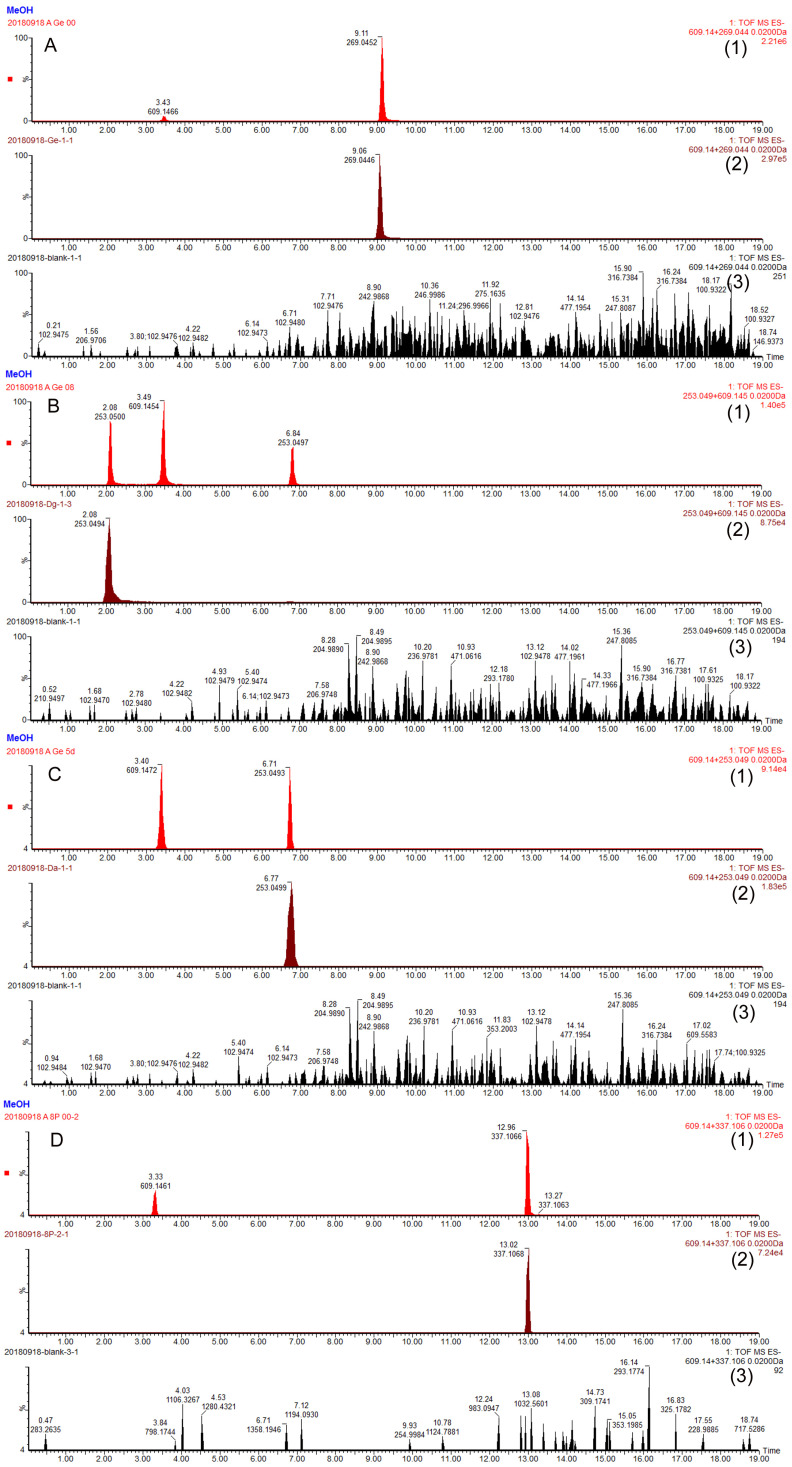
The typical chromatograms of (**A**) genistein, (**B**) dihydrogenistein and 5-hydroxy equol at 8 h, (**C**) 5-hydroxy equol on the fifth day, (**D**) 8-prenyl genistein and rutin (internal standard, IS) in anaerobic incubation of rat-gut microflora. (**1**) Genistein and 8-prenyl genistein incubated in gut microflora and spiked with IS; (**2**) blank microflora spiked with four flavonoids; (**3**) blank microflora.

**Figure 3 molecules-27-07811-f003:**
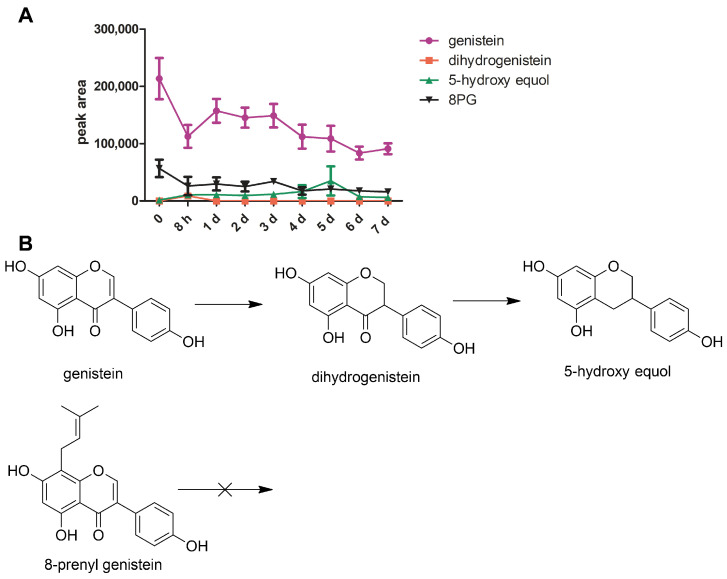
Time course of genistein and 8-prenyl genistein (8PG) conversion by rat-gut bacteria during 7 days. (**A**) Genistein (●) was transformed via dihydrogenistein (▪) to 5-hydoxy equol (▲), while no detectable metabolites of 8PG were identified; (**B**) the transformation pathway of genistein and 8PG. The symbols indicate the means of triplicate experiments. The error bars indicate standard error of mean.

**Figure 4 molecules-27-07811-f004:**
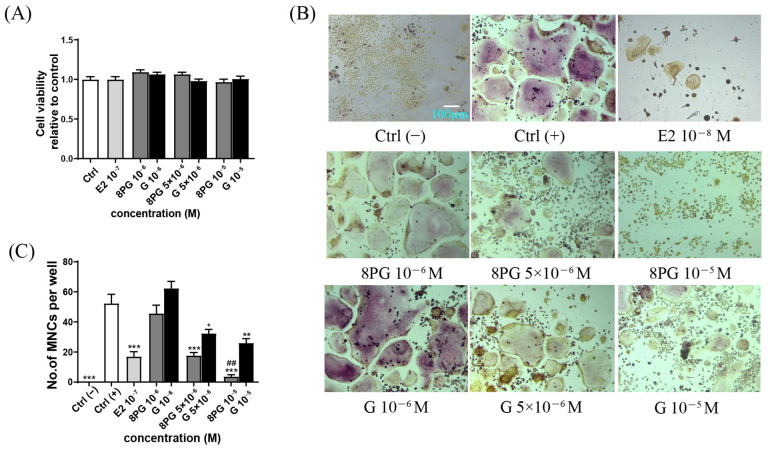
The effects of 8PG and genistein (G) on morphological changes in TRAP-positive RAW264.7 cells induced by RANKL for 4 days. (**A**) Evaluation of the cytotoxic effects of 8PG and G in RAW 264.7 pre-osteoclastic cells without RANKL induced by CCK-8 assay. (**B**) TRAP staining photographs captured under light microscope. C (−) represents control group was treated with only 1% ethanol; C (+) represents control group in which cells were induced with 1% ethanol, 100 ng/mL RANKL and 30 ng/mL M-CSF; scale bars: 200 μm. (**C**) Number of TRAP-positive multi-nucleated cells (MNCs) were counted as osteoclast cells (>3 nucleus); E2: 10^−8^ M, 17-β estradiol, as a positive control; data are means ± SEM (*n* = 3). * *p* < 0.05, ** *p* < 0.01, *** *p* < 0.001 versus C (+); ## *p* < 0.01 vs. genistein (G)-treated group at the same tested concentrations.

**Figure 5 molecules-27-07811-f005:**
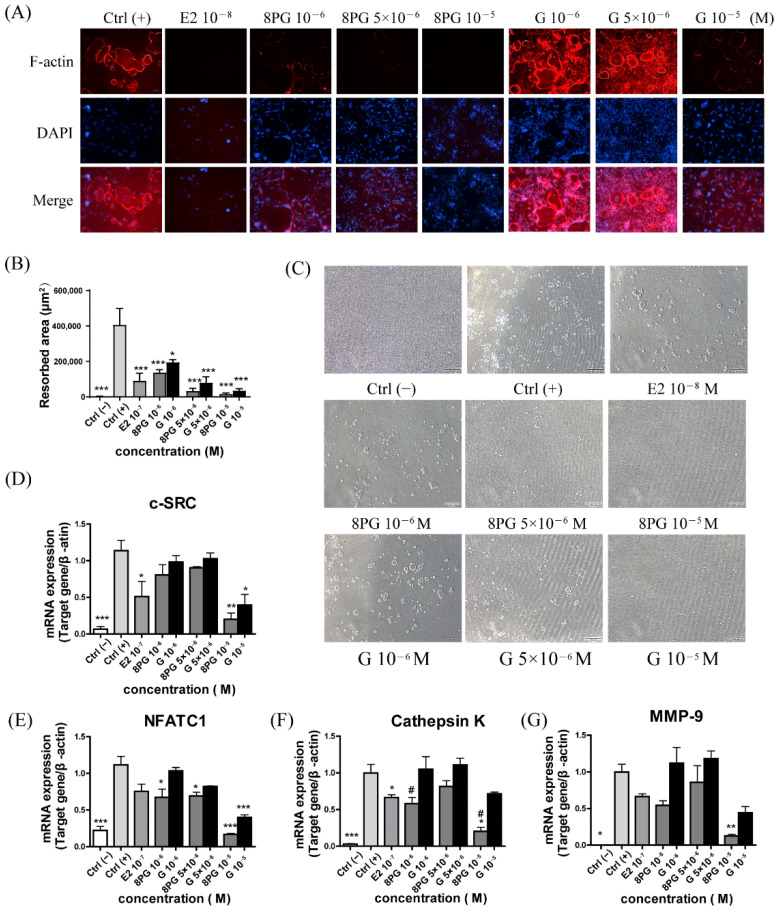
The effects of 8PG and genistein (**G**) on osteoclastic activity and the mRNA expression of osteoclast-related genes in RANKL-induced RAW264.7 cells. (**A**) Representative fluorescence images of F-actin ring formation in osteoclasts from RAW264.7 cells under osteoclast differentiation medium for 4 days. F-actin ring (red), DAPI (blue). Scale bar is 100 μm. (**B**) The total area of resorbed pits. (**C**) Representative images of bone-resorption pits. Scale bar: 100 μm; (**D**–**G**) mRNA expression of RANKL-induced osteoclast-associated genes, including c-SRC (**D**), NFATC1 (**E**), Cathepsin K (**F**) and MMP-9 (**G**). Results were expressed as mean ± SEM (*n* = 3), * *p* < 0.05, ** *p* < 0.01, *** *p* < 0.001 versus Ctrl (+); # *p* < 0.05 vs. genistein (**G**)-treated group at the same concentrations.

**Figure 6 molecules-27-07811-f006:**
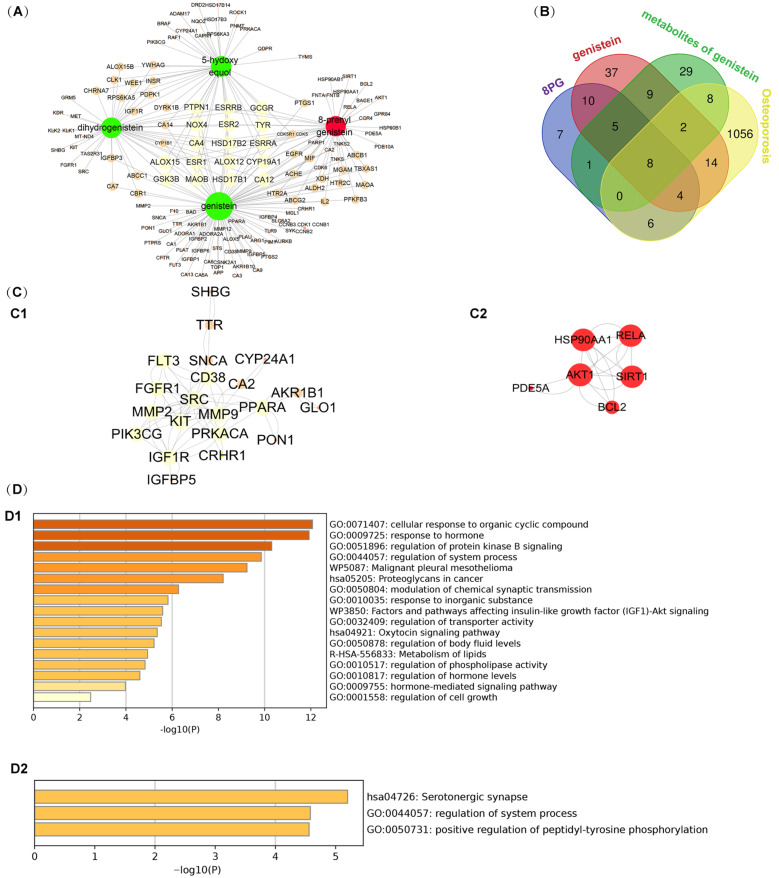
The predicted targets and pharmacological information of 8PG and genistein. (**A**) Compound-target network of 8PG, genistein, dihydrogenistein, and 5-hydoxy equol. (**B**) The overlapping targets between osteoporosis-related targets and the targets of 8PG, genistein, dihydrogenistein, and 5-hydoxy equol. (**C**) The protein–protein interactions among the overlapped targets, including genistein and its metabolites (**C1**) and 8PG (**C2**). (**D**) The enriched terms across input genes, including genistein and its metabolites (**D1**) and 8PG (**D2**).

## Data Availability

The data generated in this study are available from the corresponding author and first author upon request.

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
