# Peer review of "Genistein Modified with 8-Prenyl Group Suppresses Osteoclast Activity Directly via Its Prototype but Not Metabolite by Gut Microbiota"

_molecules, 2022, doi:10.3390/molecules27227811_

Round 1

Reviewer 1 Report

The authors present a manuscript detailing interesting results and discussion relating to the effects of genistein on osteoclast activity, and other than some grammatical issues throughout, I only suggest minor corrections:

Line 46 - the phrase 'bone remodelling-osteoclast-involved bone resorption' is awkward, suggest to reduce number of hyphens by rephrasing

Line 54 – is of great importance…

Line 57 – space before citation

Line 64 – incomplete/awkward sentence, perhaps ‘limitations given that…’

Line 68 – as reported in previous work

Line 77 – awkward, rephrase

Line 86 – check tense – behaviours were summarized.

Line 86 – remove ‘the’ preceding the reference to the figure

This whole paragraph has some grammatical issues

Figure 1 caption – repeats ‘of behaviours’

Line 106 – awkward, using in vitro incubation methods. OR an in vitro incubation method.

Line 171 – score greater than…

Line 175 – some text here is repetitive, better to merge with previous sentence

Line 182 – awkward sentence, needs rephrasing

Line 200 – unclear what is meant by ‘after a soy intake’, but probably grammatical error

Methods section seems fine.

Author Response

Reviewer 1

The authors present a manuscript detailing interesting results and discussion relating to the effects of genistein on osteoclast activity, and other than some grammatical issues throughout, I only suggest minor corrections:

Line 46 - the phrase 'bone remodelling-osteoclast-involved bone resorption' is awkward, suggest to reduce number of hyphens by rephrasing

Response: Thank you. We have revised it as “Osteoporosis is formed by the dramatic imbalance of bone remodeling, namely osteoclast-involved bone resorption outweigh osteoblast-involved bone formation”.

Line 54 – is of great importance…

Response: We have revised it as “is very important”

Line 57 – space before citation

Response: We have checked the space carefully and revised it accordingly.

Line 64 – incomplete/awkward sentence, perhaps ‘limitations given that…’

Response: We have revised it.

Line 68 – as reported in previous work

Response: We have modified it as “Our previous work demonstrated that”

Line 77 – awkward, rephrase

Response: We have revised it.

Line 86 – check tense – behaviours were summarized.

Response: We have revised it carefully.

Line 86 – remove ‘the’ preceding the reference to the figure

This whole paragraph has some grammatical issues

Response: We have modified the whole paragraph as “The mass fragmentations of 8-prenelylated genistein and genistein were listed in Fig.1A. 8-prenelylated genistein (8PG) presented a parent deprotonated ion at m/z 337.1075 [M-H]-. In the MS/MS spectrometry, it produced three primary ions at m/z 282.0528 [M-H-C4H7]-, 263.0501[M-H-C4H7-CO]-, and 225.0553 [M-H-C4H7-CO]-. Genistein showed a parent ion at m/z 269.0445 [M-H]- and two primary ions at m/z 241.0486 [M-H-CO]- and 213.0557 [M-H-2CO]- (Fig.1B). It also produced a fragment ion at m/z 225.0545[M-H-CO2]-. Interestingly, this was formed by losing O of C ring in flavones, which was reported in the previous work [17]. The fragment ions at m/z 201.0543 [M-H-C3O2]- and 159.0434[M-H-C3O2- C2H2O]- were produced by losing C3O2 and C2H2O. Meanwhile, it could also form the fragment ions by Retro Diels-Alder reaction (RDA cleavage), such as m/z 133.0279[0,3B]-”.

Figure 1 caption – repeats ‘of behaviours’

Response: It has been deleted it.

Line 106 – awkward, using in vitro incubation methods. OR an in vitro incubation method.

Response: We revised it as “using an in vitro incubation method”.

Line 171 – score greater than…

Response: We have revised it as “Score_gda > 0.1”

Line 175 – some text here is repetitive, better to merge with previous sentence

Response: We have modified the text.

Line 182 – awkward sentence, needs rephrasing

Response: The text has been revised.

Line 200 – unclear what is meant by ‘after a soy intake’, but probably grammatical error

Response: It means that only 25-60% of adults excrete equol after the consumption of soy. This sentence was modified as “However, only 25-60% of adults excrete equol after soy consumption, which is believed to be dependent on the presence of equol-forming bacteria in the gut microflora”.

Methods section seems fine.

Response: Thank you.

Author Response

Reviewer 2

The manuscript entitled “Genistein modified with 8-prenyl group suppresses osteoclast activity directly via its prototype but not metabolite by gut microbiota “ investigates the anticlastogenic activity of 8-prenyl and its effect on the intestinal biotransformation of 8PG. The introduction is well written, and the Methods are clear. However, the results section needs more attention as well as the discussion. A comparison between the human gut microbiota and the rat gut microbiota used in this study needs to be elucidated.

Response: The comparison metabolism study of genistein in human and rat gut microflora was added on page 9, line 205, as follows: “Coldham et al. [28] compared the metabolism of genistein in human and rat gut microflora demonstrating that the same metabolites of genistein were identified after incubation with rat caecal microflora and human fecal microflora, while the transformation time of these metabolites was different between the two species”

We also added a limitation in the conclusion as: “In addition, the use of rat microbiota but not human microbiota for 8PG transformation is a limitation for the present study. Further metabolism study of 8PG by human microbiota will be needed.”

  1. The paper needs more rigor to format used for units, references, concentrations, and figures quality.

Response: Thanks for your kindly suggestions, we have further polished the paper.

  1. Figure 4’ caption did not match the panels, neither the text (line 133, 134). Please be consistent with the text, MNCs not osteoclasts.

Response: we have unified all of the description as MNCs.

  1. Control (+) is missing in the cell viability test.

Response: The cell viability test aims to rule out the cytotoxicity-induced suppression effect on osteoclast differentiation by compounds, therefore, no osteoclast inducer was added to the cells when the CCK8 test was performed (MTS assay has been deleted). In order to make it more clear, Ctrl (+) has been revised to Ctrl, and the legend of figure 4 was revised accordingly.

  1. MTS is a proliferation/metabolic activity test, why this test was used?

Response: The CCK-8 test was added to evaluated the cell viability. The MTS assay was deleled. The methodology and result of “4.6.1 Cell cytotoxicity assay” and “2.3. 8PG exhibited more potent activity than genistein on suppressing RANKL-induced osteoclastogenesis, F-actin ring formation, and bone resorption activity” were revised accordingly.

  1. For what E2 does stand for?

Response: E2 stand for 17-β estradiol. We have added this information in line 132 and the legend of figure 4.

  1. Figure 5 caption did not match the panels. Panel C: images are not clear, higher magnification/resolution are needed.

Response: The Figure 5 caption has been revised and the Panel images were replaced with more clear ones.

  1. Why Relative gene expression was used instead of common delta delta Ct for gene expression by qPCR, please provide results with delta delta Ct method?

Response: We calculated the relative gene expression based on the common delta Ct method, and futher norminalized with the control group. This method was commonly used by many published papers (British Journal of Pharmacology (2018)175, 859), (FASEB J. (2019)33, 2574–2586), (Molecules (2014), 19, 18465).

  1. It is not clear how did you determine the 42 overlapped targets between 8-prenelylated genistein, 172 genistein, metabolites of genistein and osteoporosis

Response: The 42 overlapped targets were calculated as shown in the below chart. This means that the overlapped targets between 8PG and osteoporosis, overlapped targets between genistein and osteoporosis, and overlapped targets between metabolites of genistein and osteoporosis were used. Between these three overlapped targets, they also had the common targets, and duplicates were deleted. Then, 42 unique overlapped targets were used in our work. In order to describe it clearly, we have rewritten lines 181-185 of the revised manuscript.

Reviewer 3 Report

The author presented an in vitro evaluation about the potential anti-ostoclastogenesis role of genistein and derivatives. The work is well designed and presented. In vivo study will be mandatory.

i don't understand the statistical analysis and graphical presentation

for example, in fig.4B: there are two ## in the figure but three ### indicated in the legend. Again, in the legend appears ^^^p<0.001 but it is absent in the figure.....

fig.5D and E: the reference control C(+) is 1.0 or not ? seems higher. F and G is right .

8PGx10-5 in 5F and 5G have * or **. It is bizarre to me because C(-) is lower and it has *  

Again, in the legend, appears ###P<0.001 but i can't see nothing with this mark in the figure.

A re-viewing of statistics, legends and graphics is necessary

Author Response

Reviewer 3

  1. The author presented an in vitro evaluation about the potential anti-ostoclastogenesis role of genistein and derivatives. The work is well designed and presented. In vivo study will be mandatory.

Response: We agree with you that in vivo study will strengthen our results. However, the aim of the present study is to investigate the activities and mechanism of 8PG and genistein on osteoclastogenesis in RANKL-induced RAW264.7 cell model besides to study the metabolism of 8PG and genistein by gut microbiota, therefore our study focus on in vitro experiments. As we mentioned in the conclusion, “However, it should be noted that the present study only evaluated the in vitro anti-osteoclastogenic activities of 8PG, and in vivo study will be further required to confirm the role of these signaling pathways in mediating bone-sparing function of 8PG”, the in vivo study is in our next plan.

  1. I don't understand the statistical analysis and graphical presentation

for example, in fig.4B: there are two ## in the figure but three ### indicated in the legend. Again, in the legend appears ^^^p<0.001 but it is absent in the figure.....

fig.5D and E: the reference control C (+) is 1.0 or not ? seems higher. F and G is right .

8PGx10-5 in 5F and 5G have * or **. It is bizarre to me because C (-) is lower and it has *

Response: We have corrected the description of statistical analysis in the manuscript. Some detailed explination is as follows:

  • 5D and E: the reference control C(+) is 1.0 or not ? seems higher

The control is 1.0. The figures have been revised.

  • It is bizarre to me because C (-) is lower, and it has *

It should be “***”. A wrong asterisk mark was made during the layout of pictures in Adobe illustrator software. Sorry for that.

  1. Again, in the legend, appears ###P<0.001 but i can't see nothing with this mark in the figure.

Response: We have corrected it in the legends of figure 4 and 5.

  1. A re-viewing of statistics, legends and graphics is necessary

Response: We have revised them carefully. 

Round 2

Reviewer 2 Report

I would like to thank the authors for the concise revision of the paper.